

# Assessing the utility of an institutional publications officer: a pilot assessment

Kelly D. Cobey[1,2,3], James Galipeau[1], Larissa Shamseer[1,2] and David Moher[1,2]

[1] Centre for Journalology, Clinical Epidemiology Program, Ottawa Hospital Research Institute, Ottawa, Canada
[2] School of Epidemiology, Public Health, and Preventative Medicine, University of Ottawa, Ottawa, Canada
[3] School of Natural Sciences, Department of Psychology, University of Stirling, Stirling, United Kingdom

## ABSTRACT

**Background**. The scholarly publication landscape is changing rapidly. We investigated whether the introduction of an institutional publications officer might help facilitate better knowledge of publication topics and related resources, and effectively support researchers to publish.

**Methods**. In September 2015, a purpose-built survey about researchers' knowledge and perceptions of publication practices was administered at five Ottawa area research institutions. Subsequently, we publicly announced a newly hired publications officer (KDC) who then began conducting outreach at two of the institutions. Specifically, the publications officer gave presentations, held one-to-one consultations, developed electronic newsletter content, and generated and maintained a webpage of resources. In March 2016, we re-surveyed our participants regarding their knowledge and perceptions of publishing. Mean scores to the perception questions, and the percent of correct responses to the knowledge questions, pre and post survey, were computed for each item. The difference between these means or calculated percentages was then examined across the survey measures.

**Results**. 82 participants completed both surveys. Of this group, 29 indicated that they had exposure to the publications officer, while the remaining 53 indicated they did not. Interaction with the publications officer led to improvements in half of the knowledge items (7/14 variables). While improvements in knowledge of publishing were also found among those who reported not to have interacted with the publications officer (9/14), these effects were often smaller in magnitude. Scores for some publication knowledge variables actually decreased between the pre and post survey (3/14). Effects for researchers' perceptions of publishing increased for 5/6 variables in the group that interacted with the publications officer.

**Discussion**. This pilot provides initial indication that, in a short timeframe, introducing an institutional publications officer may improve knowledge and perceptions surrounding publishing. This study is limited by its modest sample size and temporal relationship between the introduction of the publications officer and changes in knowledge and perceptions. A randomized trial examining the publications officer as an effective intervention is needed.

Corresponding author
Kelly D. Cobey, kcobey@toh.on.ca, kcobey@uottawa.ca

## BACKGROUND

In 1994, Doug Altman, a world leader in biomedical research methodology, statistics, and reporting, stated that "we need less research, better research, and research done for the right reasons" (*Altman, 1994*). More than 20 years later these sentiments remain profound. Increasingly researchers have doubts about the way science gets conducted and reported. The irreproducibility of research has been highlighted as a central concern (*Baker, 2016*; *Begley & Ioannidis, 2015*; *Begley & Ellis, 2012*; *Buck, 2015*; *Collins & Tabak, 2012*; *Freedman, Cockburn & Simcoe, 2015*). This concern has been echoed in fields outside of biomedicine, including psychology (*Open Science Collaboration, 2015*). Similarly, concerns about selective reporting, publication bias, incomplete reporting, data sharing, and biased interpretation in writing (e.g., "spin"), have been expressed (*Boutron et al., 2010*; *Chan et al., 2004*; *Dwan et al., 2013*; *Glasziou et al., 2008*; *Kilkenny et al., 2009*; *Saini et al., 2014*). These problems have far reaching and multiplicative consequences: they have the potential to, directly or indirectly, delay knowledge and the discovery of novel interventions to treat or cure diseases.

Globally, there is some action. Several large funders have implemented open access and/or open data sharing policies. Open access and data sharing requirements help to ensure that research is published and that it is easily accessible, so that unnecessary duplication can be avoided and data can be used for secondary research purposes. This has the potential to enhance transparency, facilitate reproducibility, and to optimize funder investments in the research. Journals have also acknowledged problems in the conduct and reporting of biomedical research. The Lancet ran a special series in 2014 entitled *Research: increasing value, reducing waste*, which addressed this issue, and potential solutions (*Al-Shahi Salman et al., 2014*; *Chalmers et al., 2014*; *Chan et al., 2014*; *Glasziou et al., 2014*; *Ioannidis et al., 2014*). Several journals have also moved to adopt reporting guidelines —checklists of essential information to report in a manuscript—in an effort to mitigate incomplete reporting (*Shamseer et al., 2016*). Evidence suggests that endorsement and use of reporting guidelines is indeed associated with improvements in the quality of reporting (*Percie du Sert, 2011*; *Stevens et al., 2014*; *Turner et al., 2012*).

While these changes may be progressive and positive, each new policy, publication tool, or change to publication practice creates new complexities and responsibilities for researchers. These changes require time and effort from researchers if they are to understand and effectively adopt them. How are researchers expected to keep pace with these changes and ensure compliance? Another consequence of newly introduced publication policies and practices is that they may generate significant burdens for research institutions and universities who are responsible for supporting their researchers' activities and success. Recommendations for institutions to support compliance to changes in the publication landscape are plentiful. For example, in their recently adopted Statement of Principles on Digital Data Management, the Canadian Tri-Agency National Funder noted a set of seven responsibilities for institutions to support robust and open data sharing. Examples of responsibilities noted include 'promoting the importance of data management to researchers, staff and students' and 'providing their affiliated researchers with guidance to

properly manage their data in accordance with both the principles outlined (above) and research community best practices, including the development of data management plans' (*Tri-Agency Statement of Principles on Digital Data Management - Science.gc.ca., 2015*).

Who is monitoring the steps institutions are taking to provide this support? As Begley and colleagues (*2015*) recently noted, institutions may not be upholding their responsibility to provide training and resources to researchers to support the high quality, transparent, and clearly reported research that is needed to help ensure the integrity of science. Indeed, as stakeholders, institutions have been markedly absent from discussions on steps to take to improve biomedical research. One way institutions could take responsibility for supporting researchers could be through the introduction of institutional publications officers (*Moher & Altman, 2015*). Institutional publications officers could provide support to researchers at the back end of the research process. For example, they could help keep researchers up-to-date on best practices related to expectations or requirements in regards to reporting publications, such as how to find and use reporting guidelines. Publications officers could also help researchers keep pace with newly introduced open access policies and signpost them to resources such as internal repositories, or external tools like the Open Science Framework (*OSF, 2016*) (https://osf.io/). Outreach on how target a journal for submission and how to write a cover letter may be of further benefit. In addition, advice on how to navigate the peer review process, which has undergone a recent paradigm shift with the introduction of post-publication peer review, as well as changes to the openness of peer review, could be facilitated. Publications officers could work to ensure internal institutional policies related to publishing are updated to keep pace with broader international changes. Finally, they could ensure that the institutional policies and procedures acknowledge developments of novel tools such as research identifiers (i.e., ORCID) for tracking publications, and metrics and alternative metrics (e.g., views, downloads, social media uptake) for monitoring research impact.

We recently introduced a publications officer at our institution, The Ottawa Hospital Research Institute (OHRI) (*Cobey et al., 2016*). Here, we aim to describe the effect of the first six months of outreach our publications officer provided at our institution and at the neighbouring Children's Hospital of Eastern Ontario Research Institute (CHEO RI). We describe a pilot evaluation of the role's impact to date.

## METHODS

### Design

In September, 2015 we administered an online survey (via SurveyMonkey) to assess researchers' knowledge and perceptions of publishing. Researchers of all levels of seniority were invited to take part. The sample for this survey was a convenience sample administered at five Ottawa-based hospital research institutions. Specifically, surveys were sent to research staff based at OHRI and CHEO RI (experimental sites), and to researchers at three other local institutions, namely Bruyere Research Institute, The Royal Ottawa Hospital, and The University of Ottawa Heart Institute (control sites). The size of these research institutions varies considerably, with OHRI reporting more than
500 active research investigators (http://www.ottawahospital.on.ca/annualreport/fast-facts_en.html), and Bruyere Research Institute reporting fewer than 45 active investigators (http://www.bruyere.org/documents/154/AR_2016_En.pdf). All research investigators, and research staff, at each site were invited to participate. Study approval was given (Ottawa Health Science Network Research Ethics Board: 20150420-01H; The Royal Research Ethics Board: 2015018; Bruyere Research Ethics Board: M16-15-032) or waived (CHEO RI) by each location's research ethics board. Participants were initially recruited via e-mail, using an approved recruitment script which was sent to their institutional e-mail addresses from their respective administration. E-mails contained a link to our online survey. Participants provided online consent before accessing the survey.

Following this 'pre survey', on Sept 25th, 2015 hospital administration announced the new publications officer role via e-mail (performed by KDC) at OHRI and CHEO RI. The publications officer, our intervention, then provided approximately six months of active outreach at these two sites. A 'post survey', assessing publication attitudes and perceptions, was then circulated by the research team via e-mail. This e-mail was sent in March and was sent to all respondents to the 'pre survey'. Participants were told who the primary investigator was (DM) and that the purpose of the study was to examine researchers' knowledge and perceptions of publishing. Participants took part in the study voluntarily, but were informed that they would be entered into a draw to win an iPad mini after each of the two surveys (if they completed these).

## Publications officer intervention

The newly hired "Publications Officer" served as the study intervention. Our rationale for this intervention was that a localized role to support researchers in navigating the publications landscape would lead to improvements in knowledge and perception of publishing topics. As part of our rollout of this position, within the first six months the publications officer gave 24 outreach presentations across the OHRI and CHEO RI. These presentations were given face-to-face, as well as via video-conferencing software. Presentations were open to researchers of all levels of seniority at the experimental sites, and targeted both clinical and pre-clinical researchers. A typical presentation provided an overview of the newly introduced publications officer role, and highlighted internal publication related resources. A webpage of freely available resources to which researchers were sent the details of via email was also generated and updated frequently (See http://www.ohri.ca/journalology/). The publications officer was also available for one-on-one consulting, and met with 66 individuals during the study intervention period who contacted her on 94 individual occasions. Table 1 offers a summary of the topics discussed with researchers during consultations. Note the frequency of topics ($N = 79$) addressed is higher than 66 as some individuals consulted the publications officer multiple times, or about multiple distinct topics.

## Survey items

The surveys used were purpose-built for this study and also included items intended for longer term monitoring not described in this report. Researchers' institutions and email

**Table 1** Types of questions, and their frequency, received by the publications officer during her first six months of providing one-to-one consultations at OHRI and CHEO RI.

| Topic | Frequency (%) |
|---|---|
| Open Access (e.g., available funding, how to be compliant, institutional repository) | 17 (21.52) |
| Predatory Journals (e.g., how to know if a journal is predatory; what to do after submitting to a predatory journal) | 14 (17.72) |
| Submission process (e.g., where to submit, how to select a journal, help with cover letter) | 13 (16.46) |
| Writing (e.g., use of reporting guidelines, feedback on writing, available writing tools) | 12 (15.19) |
| Peer Review (e.g., responding to reviewers, making sense of reviewer comments) | 6 (7.59) |
| Publication Ethics (e.g., duplicate publications, copyright, plagiarism) | 5 (6.33) |
| Authorship (e.g., authorship disputes, who qualifies for authorship) | 4 (5.06) |
| Other (e.g., remit of publications officer role; ORCID identifier) | 8 (10.13) |

addresses were collected on the first survey so that we would be able to re-contact them to complete the second survey. Participants were asked to respond to 14 multiple choice survey items designed to assess their knowledge of journalology (i.e., publication science) topics. As an example, one item asked participants 'What is Creative Commons?' and another 'What are reporting guidelines?' For a full description of knowledge questions asked and possible responses, please see Appendix 1. These questions capture knowledge of a range of potentially relevant journalology resources, platforms, and terms. While they may not assess all relevant domains of knowledge, we selected these items as we felt they broadly represented distinct domains of publication science. In addition to these items, participants were asked to respond to 6 items designed to measure their perceptions and intentions related to publishing (See Fig. 1). Participants responded to these items on a Likert scale of one to seven, with endpoints 'Not at all true' and 'Completely true', respectively. An example item is 'I am confident in my understanding of publication ethics'.

## Data analysis

All data was stored securely and de-identified prior to analysis. We provide descriptive summary data for those who did and did not interact with the publications officer. The journalology knowledge questions, which were recoded to be dichotomous variables (i.e., participants' answers were correct or incorrect), were summarized as proportions and percentages. The publications perceptions items, which were continuous variables, were summarized by means and standard errors. We compared changes across the pre and post survey between each group.

(a)

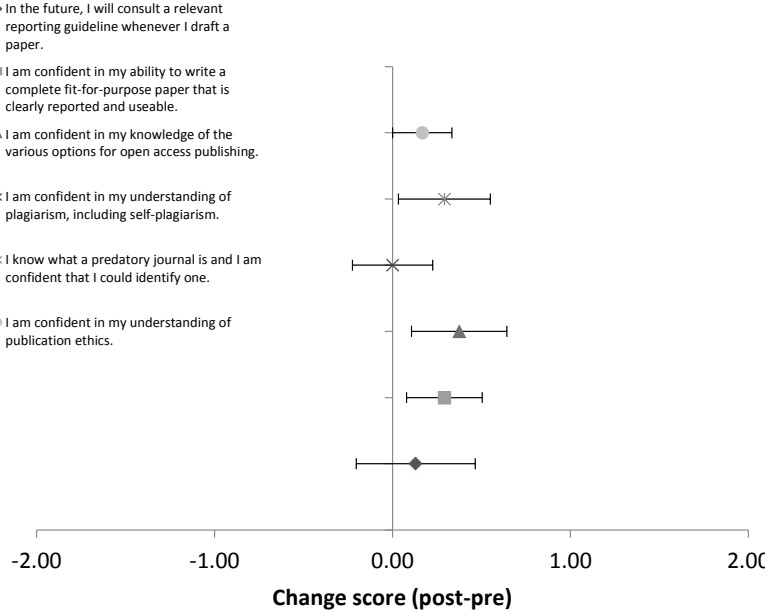

(b)

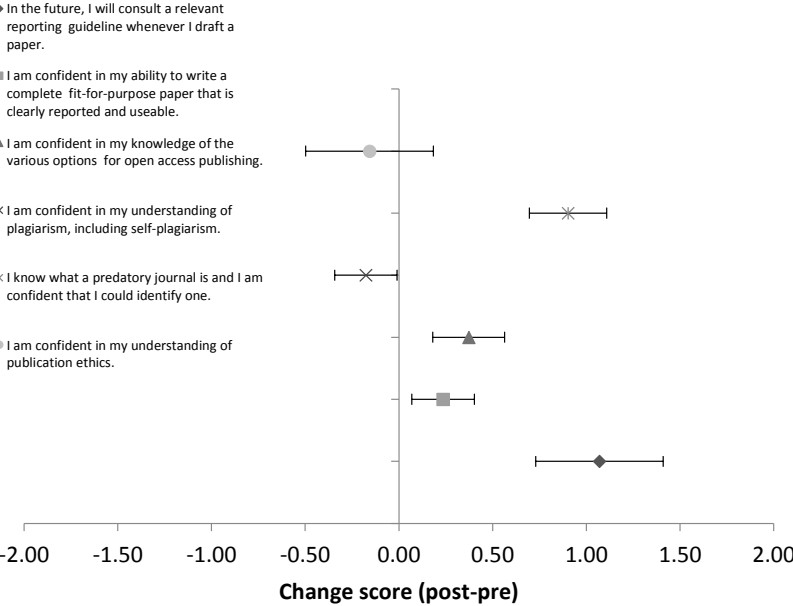

**Figure 1** Mean (±SE) change in publication perceptions between the post and pre survey for participants who did (A), and did not (B), interact with the publications officer (PO).

## Participants

A total of 119 participants completed the first survey; however, six provided emails that were no longer in service at the time of the post-survey, and 31 failed to complete the follow-up survey. Therefore, participants analyzed were 82 individuals ($N = 41$ male, $N = 40$ female, $N = 1$ missing data) based at OHRI, CHEO RI, Bruyere Research Institute, The Royal Ottawa Hospital, and the University of Ottawa Heart Institute.

## Modifications from protocol

While we had initially hoped to compare responses at sites receiving our publications officer intervention (OHRI and CHEO RI) with those that did not, the modest response rate made this unfeasible. We therefore compared participants who explicitly indicated that they had interacted with the publications officer to those who indicated they had not interacted with the publications officer for each of our variables of interest. Specifically, on the second survey we asked participants to indicate if they had: (1) visited the Centre for Journalology website maintained by the publications officer; (2) received an email or newsletter from the publications officer; (3) attended a seminar held by the publications officer; (4) had a one-to-one meeting with the publications officer; or (5) had any other interaction with the publications officer. If participants indicated yes to any of these five options, they were considered to have interacted with the publications officer ($N = 29$). Those that indicated they had not used any services were classified as not having interacted with the publications officer ($N = 53$).

## RESULTS

The proportions of correct responses to the publication knowledge questions posed during the pre and post survey, for those who did and did not interact with the publications officer, are summarized in Table 2. While neither group had exposure to the publications officer prior to the first survey, there were differences in baseline responses between the groups. For 12/14 variables, participants who went on to interact with the publications officer had higher scores at baseline. For 13 out of the 14 variables, the proportion of correct responses was higher at the time of the post survey for the group who interacted with the publications officer.

Table 3 summarizes the change in percentage of correct responses to each publication knowledge variable from the pre to the post survey. In general, publication knowledge tended to increase from the pre to the post survey irrespective of whether participants interacted with the publications officer or not. Participants who interacted with the publications officer improved their scores from the pre to the post-survey for 7/14 variables. This finding is in spite of the fact that this group tended to have greater baseline knowledge, meaning they had less room for improvement. Furthermore, for one variable where participants were asked what a redundant publication was, those who interacted with the publications officer were 100% correct leaving no potential room for improvement.

In a few notable cases, exposure to the publications officer resulted in decreases in correct response percentages during the post survey ($N = 3/14$). For example, with the item 'How Is a journal's impact factor calculated?' participants who interacted with the publications

**Table 2** **Proportion and percent correct of responses to the publication knowledge questions by study group for the pre and post survey measures.** The rightmost column indicates whether the group who interacted with the publications officer (PO) has a higher post score.

| Survey Question | | Did not interact with PO | | | Interacted with PO | | | Difference in post score |
|---|---|---|---|---|---|---|---|---|
| | | N | Frequency Correct | % Correct | N | Frequency Correct | % Correct | |
| What is Journalology? | Pre | 53 | 27 | 50.94 | 21 | 15 | 71.43 | ✓ |
| | Post | 53 | 27 | 50.94 | 21 | 15 | 71.43 | 20.49 |
| Enabling free access to a research publication, for instance, through an institutional repository, is often referred to as: | Pre | 47 | 5 | 10.64 | 21 | 4 | 19.05 | ✓ |
| | Post | 47 | 5 | 10.64 | 21 | 10 | 47.62 | 36.98 |
| What is Creative Commons? | Pre | 47 | 9 | 19.15 | 21 | 7 | 33.33 | ✓ |
| | Post | 47 | 13 | 27.66 | 21 | 8 | 38.10 | 10.44 |
| Which of the following is true of open access publications? | Pre | 47 | 12 | 25.53 | 20 | 9 | 45.00 | ✓ |
| | Post | 47 | 17 | 36.17 | 20 | 11 | 55.00 | 18.83 |
| How is a journal's impact factor calculated? | Pre | 46 | 13 | 28.26 | 21 | 15 | 71.43 | ✓ |
| | Post | 46 | 17 | 36.96 | 21 | 9 | 42.86 | 5.90 |
| **Approximately how much money is estimated to be wasted annually, globally, in health research?** | Pre | 42 | 5 | 11.90 | 20 | 3 | 15.00 | ✓ |
| | Post | 42 | 6 | 14.29 | 20 | 8 | 40.00 | 27.71 |
| Roughly what percent of biomedical conference presentations are subsequently published as full length research articles? | Pre | 43 | 26 | 60.47 | 21 | 11 | 52.38 | ✓ |
| | Post | 43 | 28 | 65.12 | 21 | 15 | 71.43 | 7.31 |
| What are reporting guidelines? | Pre | 45 | 38 | 84.44 | 21 | 19 | 90.48 | ✓ |
| | Post | 45 | 38 | 84.44 | 21 | 21 | 100.00 | 15.56 |
| **Which of the following is *always* true of predatory journals?** | Pre | 42 | 19 | 45.24 | 19 | 8 | 42.11 | ✓ |
| | Post | 42 | 22 | 52.38 | 19 | 12 | 63.16 | 10.78 |
| Which of these is not an example of publications bias? | Pre | 39 | 24 | 61.54 | 20 | 15 | 75.00 | ✓ |
| | Post | 39 | 26 | 66.67 | 20 | 15 | 75.00 | 8.33 |
| Which one(s) of these impact factors includes all articles indexed in the Web of Science? | Pre | 46 | 11 | 23.91 | 21 | 9 | 42.86 | ✓ |
| | Post | 46 | 15 | 32.61 | 21 | 9 | 42.86 | 10.25 |
| When findings from a research study do not agree with your initial hypothesis, it is acceptable/recommended to | Pre | 45 | 33 | 73.33 | 20 | 17 | 85.00 | $\chi =$ |
| | Post | 45 | 30 | 66.67 | 20 | 12 | 60.00 | $-6.67$ |
| Reporting guidelines are useful for (check all that apply): | Pre | 39 | 16 | 41.03 | 19 | 12 | 63.16 | ✓ |
| | Post | 39 | 16 | 41.03 | 19 | 10 | 52.63 | 11.60 |
| A redundant publication is: | Pre | 44 | 25 | 56.82 | 19 | 19 | 100.00 | ✓ |
| | Post | 44 | 35 | 79.55 | 19 | 19 | 100.00 | 20.45 |

**Table 3  Difference in percentage of correct responses (post –pre survey) for each group.**

| Survey Question | Did not interact with PO Change in % correct (Post-Pre) | Interacted with PO Change in % correct (Post-Pre) |
|---|---|---|
| What is Journalology? | 0 | 0 |
| Enabling free access to a research publication, for instance, through an institutional repository, is often referred to as: | 0 | 28.57 |
| What is Creative Commons? | 8.51 | 4.76 |
| Which of the following is true of open access publications? | 10.63 | 10 |
| How is a journal's impact factor calculated? | 8.70 | −28.57 |
| **Approximately how much money is estimated to be wasted annually, globally, in health research?** | 2.38 | 25.00 |
| Roughly what percent of biomedical conference presentations are subsequently published as full length research articles? | 4.65 | 19.05 |
| What are reporting guidelines? | 0 | 9.52[a] |
| **Which of the following is *always* true of predatory journals?** | 7.12 | 21.05 |
| Which of these is not an example of publications bias? | 5.13 | 0 |
| Which one(s) of these impact factors includes all articles indexed in the Web of Science? | 8.70 | 0 |
| When findings from a research study do not agree with your initial hypothesis, it is acceptable/recommended to | −6.67 | −25.00 |
| Reporting guidelines are useful for (check all that apply): | 0 | −10.53 |
| A redundant publication is: | 22.73 | 0[a] |

**Notes.**
[a]indicates 100% on post survey.

officer responded 71.43% correct to the pre-survey, but only responded 42.86% correct to the post survey. On this same item, participants who did not interact with the publications officer improved their knowledge score by 8.70% from the pre to the post survey; however, in spite of this, knowledge on the post-survey (36.96%) nonetheless remained below the levels found among participants who interacted with the publications officer (42.86%).

Figure 1 shows the mean values for the publication perception items for each group. Mean values to responses to these items ranged from 3.86 to 6.13 (Table 4). As with the publication knowledge questions, scores across items for the group that interacted with the publications officer were higher at baseline (6/6 variables). Those participants who interacted with the publications officer tended to increase scores from the pre to the post survey (5/6 variables), and had higher post scores on most variables (5/6), despite the fact that the change in mean scores for those who did not interact with the publications officer was greater for two variables.

## DISCUSSION

One way institutions may be able to support their researchers in staying current with changes in the publication landscape is through the introduction of institutional publications

Cobey et al. (2017), *PeerJ*, DOI 10.7717/peerj.3294

**Table 4** Mean scores for responses for the publication perception items for participants who did, and who did not, interact with the publications officer (PO).

| Question | | Did not interact with PO | | | Interacted with PO | | |
|---|---|---|---|---|---|---|---|
| | | N | Mean | SD | N | Mean | SD |
| In the future, I will consult a relevant reporting guideline whenever I draft a paper. | Pre | 43 | 4.81 | 2.14 | 23 | 5.48 | 1.47 |
| | Post | 43 | 5.88 | 1.38 | 23 | 5.61 | 1.90 |
| I am confident in my ability to write a complete fit-for-purpose paper that is clearly reported and useable. | Pre | 51 | 4.75 | 1.90 | 24 | 5.33 | 1.46 |
| | Post | 51 | 4.98 | 1.68 | 24 | 5.62 | .88 |
| I am confident in my knowledge of the various options for open access publishing. | Pre | 51 | 3.86 | 1.96 | 24 | 4.46 | 1.53 |
| | Post | 51 | 4.24 | 1.78 | 24 | 4.83 | 1.49 |
| I am confident in my understanding of plagiarism, including self-plagiarism. | Pre | 51 | 5.53 | 1.22 | 24 | 6.13 | .95 |
| | Post | 51 | 5.35 | 1.37 | 24 | 6.13 | .74 |
| I know what a predatory journal is and I am confident that I could identify one. | Pre | 51 | 3.92 | 1.93 | 24 | 5.58 | 1.67 |
| | Post | 51 | 4.82 | 1.76 | 24 | 5.88 | 1.23 |
| I am confident in my understanding of publication ethics. | Pre | 51 | 5.33 | 1.44 | 24 | 5.79 | 1.02 |
| | Post | 51 | 5.18 | 1.51 | 24 | 5.96 | .95 |

officers. Here, we describe a pilot evaluation of a newly introduced publications officer role. Anecdotally, the role appears to have been positively received. This positive reception is reflected in the rapid uptake and overall number of one-to-one consultations, as well as researcher attendance and feedback at seminars. This experience suggests that the role filled a previously existing gap in services that researchers were eager to immediately address.

Our pilot findings pertaining to the effectiveness of the publications officer as a meaningful intervention provide initial empirical evidence of the potential value of this role. For seven out of the 14 variables used to assess publication knowledge, we found that researchers had higher scores after interaction with the publications officer as compared to the control. This result occurred in spite of the fact that those who interacted with the publications officer had higher baseline scores for a number of variables. One interpretation of these findings could be that those who were already interested in journalology (as evidenced by the higher baseline scores) were able to access resources previously unavailable (or unknown) to them and, in the process, increased their knowledge. This could indicate the value of a publications officer for researchers who are already knowledgeable in journalology-related topics, not only for those who are novices in this domain.

For three of the journalology knowledge items, participants who interacted with the publications officer actually decreased their scores from the pre to the post survey. It is not immediately clear why their knowledge scores decreased. However, it is worthwhile noting that in spite of these decreases, the absolute post scores were still higher in the group that interacted with the publications officer. These decreases may reflect random variation due to our small sample size. Concerning participants' perceptions of publishing, among those that interacted with the publications officer, scores tended to improve. Findings for those who did not interact with the publications officer were more inconsistent, with scores on some variables improving quite considerably, and others reducing. It is worthwhile to note that many of the mean values for responses to these items in both groups, even at the post survey measures, were below 5.5. Given the inherent importance of many of these concepts in order to publish according to best practice, the relatively low confidence rates in perceptions related to publishing is troubling. Shifting perceptions, in contrast to shifting knowledge of particular facts, may require longer periods of time to achieve robust impact.

This study is not without limitations. Firstly, our sample size was modest and underpowered to consider use of inferential testing of many hypotheses. A limitation of this work is that we failed to employ a randomized design. As a consequence, and as suggested by the baseline differences in knowledge scores we obtained, it may be that there was a selection bias such that participants who knew more about journalology subsequently were more likely to seek out and interact with the publications officer. Failure to randomly assign participants to interact with the publications officer limits our ability to draw causal inferences. Future work using a larger pool of participants' with random assignment is therefore warranted. In addition, it is difficult to know whether any effects of the publications officer intervention carried over into the control group. Given the close proximity of researchers (all based in Ottawa), this is certainly possible and may explain the increases in knowledge observed in the control group. There are known collaborations between the various sites. It is possible, for example, that those in the control group

actually did have exposure to outreach services by the publications officer, especially the webpage and electronic newsletters which were widely distributed at OHRI and CHEO RI, but that they did not recognize that these explicitly related to the publications officer when surveyed. Alternatively, some may also have indirectly gained knowledge from having interactions with colleagues who had exposure to the publications officer. A further potential confounder of our study is that participants may have adjusted their behaviour or knowledge based on the fact that they knew they were being measured/observed. As noted above, a randomized and blinded trial could address this in the future.

Finally, outreach material and presentations given by the publications officer were not all specifically developed to address each of the knowledge based questions used herein. Nor will our purpose-built knowledge based questions completely capture all knowledge areas or domains of publications science. Future studies may wish to build on the items we included, and include items that address situational knowledge. It will be important to determine how effective the various types of outreach provided by the publications officer are at increasing knowledge and strengthening perceptions in future evaluations. This will allow the services of the publications officer to be specified over time to become most effective. An in-depth evaluation of the degree and quality of interaction participants had with the publications officer was also not conducted as part of this pilot study but could prove valuable. This could help to ensure that differences observed between the control and treatment groups herein, are robust, and that improvements in publication science knowledge and perceptions are indeed a consequence of the publications officer as opposed to other factors, such as increased media coverage of these issues.

Writing a high-quality transparent manuscript, navigating through the journal submission and peer review process, and eventually publishing are important components of the research continuum. Ensuring that researchers have internal resources available to them to make sure that they are adhering to best practices and compliant with any relevant publishing policies is essential to upholding scientific integrity. Since starting in the role, our Publications Officer has engaged with senior administration locally. This engagement has led to a refresh of three institutional policies (Authorship Guideline, Data Sharing Guideline, and Publication Guideline) and discussions about how the role can provide novel insights or services for the institution. For example, in response to the development of an automated TrialsTracker tool (*Powell-Smith & Goldacre, 2016*), the publications officer is now establishing an internal audit program at OHRI to help ensure that clinical trials registered on clinicaltrials.gov, which are completed have their results publicly reported. The publications officer role may be an efficient and relatively inexpensive resource that institutions can implement to add value to, and ensure the quality of, their publications. Further research on the role and its impact, addressing the design limitation noted herein, is warranted in order to clarify and improve the impact of the publications officer positions.

## APPENDIX 1. JOURNALOLOGY KNOWLEDGE

*Participants responded to the items below to assess their knowledge of journalology. The same items were used during the pre and post survey.*

(1) What is Journalology?
   (a) The study of scientific journalism
   (b) **The study of scientific publication**
   (c) The study of journalists
   (d) The study of journals
   (e) The study of open access
   (f) None of these options
   (g) Other, please specify ___________
   (h) Don't know

(2) Enabling free access to a research publication, for instance, through an institutional repository, is often referred to as:
   (a) Blue open access
   (b) **Green open access**
   (c) Platinum open access
   (d) Hybrid open access
   (e) Don't know
   (f) Other, please specify ____________

(3) What is Creative Commons?
   (a) An open-access journal
   (b) A website where people can share their work
   (c) **An organization offering copyright licences**
   (d) A computer program that allows authors work collaboratively on a paper
   (e) Other, please specify ____________
   (f) None of the above

(4) Which of the following is true of open access publications?
   (a) The author always retains copyright
   (b) The publisher always retains copyright
   (c) Open access journals are more likely than subscription journals to allow authors to retain copyright
   (d) **Open access journals are less likely than subscription journals to allow authors to retain copyright**
   (e) None of the above

(5) How is a journal's impact factor calculated?
   (a) **It is the average number of citations to recent articles published in a particular journal in the past 2 years.**
   (b) It is the average number of times articles published in a journal have been cited in the past two years, excluding papers which have not been cited at all
   (c) It is the average number of times articles published in a journal have been cited in the past two years, excluding self-citations
   (d) It is the average number of times articles published in a journal have been cited in the past two years, excluding self-citations and papers which have not been cited at all
   (e) Other, please specify ________

(f) None of the above

(6) Approximately how much money is estimated to be wasted annually, globally, in health research?

   (a) 5 Billion
   (b) 50 Billion
   (c) **200 Billion**
   (d) 500 Billion
   (e) Other, please specify _______
   (f) There is no estimate of waste

(7) Roughly what percent of biomedical conference presentations are subsequently published as full length research articles?

   (a) 90%
   (b) 80%
   (c) 70%
   (d) 60%
   (e) **50%**
   (f) Other, please specify
   (g) There is no estimate

(8) What are reporting guidelines?

   (a) Guidance for reporters who cover health research
   (b) Guidance for researchers conducting a research study
   (c) **Guidance for authors writing up reports of their research**
   (d) Guidance for editors on how to run a journal
   (e) Other, please specify _____________
   (f) None of the above

(9) Which of the following is *always* true of predatory journals?

   (a) They don't host online submission platforms
   (b) They don't peer review
   (c) They never actually 'publish' papers
   (d) **They collect money from authors**
   (e) All of the above

(10) Which of these is not an example of publications bias?

   (a) Publishing only the results that are in line with your predictions
   (b) Failing to publish results from a study that had no statistically significant results
   (c) Omission of some study results to send a focused message
   (d) Failing to publish a study's results
   (e) **None of the above**

(11) Which one(s) of these impact factors includes all articles indexed in the Web of Science?

   (a) **Thomson Reuters Journal Citation Reports**
   (b) Global Impact Factor
   (c) Universal Impact Factor
   (d) Index Copernicus Value
   (e) Other, please specify _______

     (f) None of the above

(12) When findings from a research study do not agree with your initial hypothesis, it is acceptable/recommended to:

     (a) Collect more data before attempting to publish

     (b) Publish only the agreeable findings in order to stay focused on what's most important from the study

     (c) Modify the results so that the findings are favourable and a journal will publish them

     (d) Publish all the data, but write the discussion in a way that makes the negative findings not look so bad, so that people will still see the benefits.

     (e) **Double check that the design and analyses performed were sound and, if so, proceed with publication**

     (f) Don't bother with publication

     (g) Other, please specify _______________

     (h) None of the above

(13) Reporting guidelines are useful for (check all that apply):

     (a) Designing participant consent forms

     (b) **Writing a manuscript for consideration for publication**

     (c) **Conducting peer reviews of manuscripts**

     (d) **Decision-making by journal editors (acceptance/rejection of a manuscript)**

     (e) Interviews with reporters when discussing one's research

     (f) The media when reporting on new research

     (g) Other, please specify _____________

     (h) They are not useful

(14) A redundant publication:

     (a) Is the copying of ideas from another source

     (b) Is a novel replication of a previously published result

     (c) **Is a publication which is identical to or overlaps substantially with another publication**

     (d) Is a publication which fails to declare conflict of interest

     (e) All of the above

### Funding

David Moher is funded through a University Research Chair, University of Ottawa. The funders had no role in study design, data collection and analysis, decision to publish, or preparation of the manuscript.

### Grant Disclosures

The following grant information was disclosed by the authors:
University Research Chair, University of Ottawa.

## Competing Interests

Kelly D. Cobey is the publications officer at the Ottawa Hospital Research Institute. No other competing interests exist.

## Author Contributions

- Kelly D. Cobey conceived and designed the experiments, performed the experiments, analyzed the data, wrote the paper, prepared figures and/or tables, reviewed drafts of the paper.
- James Galipeau, Larissa Shamseer and David Moher conceived and designed the experiments, performed the experiments, reviewed drafts of the paper.

## Human Ethics

The following information was supplied relating to ethical approvals (i.e., approving body and any reference numbers):

Ottawa Health Science Network Research Ethics Board: 20150420-01H; The Royal Research Ethics Board: 2015018; Bruyere Research Ethics Board: M16-15-032.

## Data Availability

The raw data has been supplied as Supplemental Information 1.

## Supplemental Information

Supplemental information for this article can be found online at http://dx.doi.org/10.7717/peerj.3294#supplemental-information.

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
