# Peer review of "Assessing the utility of an institutional publications officer: a pilot assessment"

_PeerJ, doi:10.7717/peerj.3294_

## Round 0.1 · original submission · Minor Revisions

I think your manuscript will be of interest to many people. To make is suitable for a broader audience, please make sure you clarify institutional details and expand your discussion about study limitations. Please address the reviewers comments for each category.

·

Basic reporting

It would be helpful to provide the following information to the reader, which appears to be missing:

- Size of the institutions (# of researchers)? ?
- How many participants were invited (i.e. what was the response rate)?
- How many participants dropped out between the pre – and the post intervention query?
- Can the use of the webresources be quantified (# of page views, downloads)?

Experimental design

The study has a number of weaknesses, the biggest of which are certainly are its small size and lack of randomization. However, these weaknesses are exposed and discussed by the authors, while the current study can lay the foundation for a larger, randomized trial.

Validity of the findings

General: I have to admit that I do not understand the rationale behind several of the questions, in fact I doubt that some of them are relevant to what the authors want to achieve/measure. For example, is it important to know what Journalology mean? Does Journalology even exist? Question 11 about Thomson Reuters? Are you aiming to improve this type of knowledge through a publication officer? Same for question 6 and 7. My point is: An increase in correct answers to these questions does not reflect what one would want to achieve through the activities of the officer. But why then use them as a measure of the efficacy of the measure?

Consider discussing the following issues:
- Potential confounder: Hawthorne effect
- General increase in knowledge about publishing throughout the study period due to increased coverage in media, at conferences, etc. (‘shift in the baseline of all participants’)
- Could it be that the effects seen (including some of the bizarre one, like the decrease in knowledge about the IF in the verum group) could be the result of random fluctuations? After all, the sample size is very small, and many variables are studied?

Additional comments

This is an interesting and relevant pilot study investigating the impact of establishing an institutional publications officer at several academic institutions in Canada. Amidst major changes in the publishing industry and the way academics report their research this is a very timely initative and investigation. The authors should be commended not only for implementing such a measure to improve the quality of research at their institution, but also for doing this in a scientific manner, i.e. complementing it with meta-research in the form of an ‘interventional trial’.
Overall I find this a very important contribution which will be of interest to many individuals and institutions. I recommend publication with minor revisions.

I sign my reviews. Ulrich Dirnagl

·

Basic reporting

It would be more meaningful to place the relevant institutions in brackets after each of the two ethical outcomes i.e. given (institution(s)) or waived (institution(s))

Typically I would expect the description of the Publications Intervention to precede details of the Survey questions – the reader can thus judge whether the questions asked fitted the intervention, rather than the reverse

“In 1994, Doug Altman stated that “we need less research, better research, and research done for the right reasons” (Altman, 1994).” Either this assumes that every reader will know who Doug Altman is (I actually do know him well!), or, if they don’t this sentence adds little beyond the information contained in the in-text reference. I suggest that you add a brief clause that places Doug in an appropriate context e.g. “one of the world’s leading experts in health research methodology, statistics, and reporting” (according to his BMJ Lifetime Achievement Award citation at https://doi.org/10.1136/bmj.b5542 ) or “subsequently, founder member of the EQUATOR Network on guidelines for health reporting” etc.

“are still profound” Prefer “remain profound”

It seems strange to have a colloquial term such as “spin” at the end of a list of technical terms. Perhaps bridge this by using the technical term with spin in brackets e.g. “biased interpretation in writing (“spin”)”.

You can’t really “delay knowledge” Perhaps “delay the spread of knowledge”?

“who are responsible for supporting their researchers’ success” – actually the institutions are responsible for supporting the activities of their researchers, not necessarily only their success!

Although I don’t like the expression of “back end of the research process” I can understand how it might be used. However this argument is confused by the following sentence which refers to both the research design and reporting stages which are at contrasting ends along the research pathway.

“Outreach on how to select a journal to submit to and how to write a cover letter may be of further benefit.” “Submit to” is cumbersome Suggest: “Outreach on how to target a journal for submission and how to write a cover letter may be of further benefit”.

“Finally, they could ensure that the institution itself is kept current on tools such as research identifiers” – I don’t see how you can ensure that an institution is kept current separate from the staff. Phrase this in terms of either the staff within the organisation or more likely I suspect, “ensure that institutional policies and procedures acknowledge developments in tools such as…..”

Experimental design

It might be helpful for the authors to review their description of the Publications Officer Intervention against the TIDIER reporting format using the template to check that they have shared as much information appropriate to replication as possible within reporting constraints. (Hoffmann TC, Glasziou PP, Boutron I, Milne R, Perera R, Moher D, Altman DG, Barbour V, Macdonald H, Johnston M, Lamb SE. Better reporting of interventions: template for intervention description and replication (TIDieR) checklist and guide. Bmj. 2014 Mar 7;348:g1687.)

If I have a difficulty with this study then this relates to the items on the survey instrument – the items confound factual statements that test knowledge e.g. total of research waste and percentage of conference presentations published with more instrumental knowledge that might usefully change researcher practice. A publication officer’s role in the institution is not in promoting journalology but in having an impact on researcher practice. However, in the absence of validated tools to assess specific knowledge the tool seems broadly fit for purpose. Perhaps future evaluation instruments could factor in more of an If…Then structure relating to behaviours e.g. “If I choose to publish with a predatory publisher Then…..” “If I don’t follow appropriate reporting guidelines Then…” In a sense, question 12 is the most useful as being closest to this scenario based evaluation format.

Validity of the findings

This case study indicates proof of concept although it is understandably circumspect when self-critiquing its methods and potential conclusions. The degree of evaluation is proportionate, and within that context, robust.

Additional comments

It is interesting to see different levels of knowledge at baseline suggesting that those who interacted with the publications officer might either have greater motivation to do so or operate in roles where that interaction is a natural expectation of their role. This makes the greater challenge that of making researchers aware of their unexpressed “need” to access the publications officer.

No action required - although it might be helpful to discuss this further challenge in the Discussion

---

## Round 0.2 · accepted · Accept

Thank you for addressing all the suggested changes. The added details and modified paragraphs have strengthened the discussion section. The overall article is straightforward and easy to read, and of great interest to researchers and institutions.